# Chronic Myeloid Leukemia in a Patient with Previous Idiopathic Thrombocytopenic Purpura: How to Manage Imatinib Together with Eltrombopag

**DOI:** 10.3390/medicina57121326

**Published:** 2021-12-03

**Authors:** Francesco Autore, Federica Sora’, Patrizia Chiusolo, Gessica Minnella, Maria Colangelo, Elena Rossi, Simona Sica

**Affiliations:** 1Dipartimento di Diagnostica per Immagini, Radioterapia Oncologica ed Ematologia, Fondazione Policlinico Universitario A. Gemelli IRCCS, 00168 Roma, Italy; federica.sora@policlinicogemelli.it (F.S.); patrizia.chiusolo@unicatt.it (P.C.); gessica.minnella@policlinicogemelli.it (G.M.); elena.rossi@unicatt.it (E.R.); simona.sica@unicatt.it (S.S.); 2Sezione di Ematologia, Dipartimento di Scienze Radiologiche ed Ematologiche, Università Cattolica del Sacro Cuore, 00168 Roma, Italy; 3UOC Genetica Medica, Fondazione Policlinico Universitario A. Gemelli IRCCS, 00168 Roma, Italy; maria.colangelo@policlinicogemelli.it

**Keywords:** chronic myeloid-leukemia, idiopathic thrombocytopenic purpura, eltrombopag, imatinib

## Abstract

The occurrence of chronic myeloid leukemia (CML), or other myeloproliferative diseases, after the development of idiopathic thrombocytopenic purpura (ITP) is very rare in the current medical literature. Considering the advances in ITP management, and the wide use of new drugs for ITP and CML, we report an unusual case with this association. Our case report focused on a 64-year-old man with long-standing ITP treated with eltrombopag, who developed hyperleukocytosis during follow-up; after specific laboratory exams, it was diagnosed as CML and he began treatment with imatinib. The treatment with eltrombopag was balanced with imatinib to stabilize his platelet count. Data on bcr-abl and JAK2 transcripts were collected and revealed an optimal response with the achievement of negativization of both molecular signatures. We could demonstrate that treatment with imatinib and eltrombopag was well tolerated and allowed complete molecular remission of CML to be achieved, as well as of ITP.

## 1. Introduction

The association of hematologic malignancies with autoimmune disorders has been well recognized; in the hematological field, idiopathic thrombocytopenic purpura (ITP) was also associated with hematological malignancies and clonal myeloid/lymphoid disorders [1]. In particular, recent papers studied patho-biological mechanisms linking ITP and clonal disorders, and their temporal association. Many published experiences focused on ITP and myelodysplastic syndrome (MDS)/chronic myelomonocytic leukemia (CMML), but when analyzing the temporal association, it was evident that, in the majority of cases, ITP followed the diagnosis of MDS/CMML and only very few cases were diagnosed as MDS/CMML after ITP [2,3,4]. 

From our clinical practice on patients affected by chronic myeloid leukemia (CML), we noted the coexistence of CML and ITP in one of our patients; when reviewing the literature, we found that it is very rare to diagnose CML or other myeloproliferative diseases after the development of ITP. Here, we describe the case of our patient diagnosed with CML, who was already treated for ITP.

## 2. Case Report

A 64-year-old male was followed in our center for ITP diagnosed in 2002. His past medical history was remarkable for arterial hypertension, requiring treatment, and he had a peptic ulcer; there was no familiarity for hematological diseases.

At ITP onset, he showed severe thrombocytopenia; the increase in megakaryocytes in the bone marrow (BM) and the negativity of other tests, including the autoimmunity profile, allowed the diagnosis of ITP. He was treated with steroids, a first line with dexamethasone and a second line with prednisone, enabling only a partial response to be reached. Because of steroid dependency and multiple relapses, he started eltrombopag in September 2011, initially at the dose of 50 mg/day, then reduced to 25 mg/day. In May 2017, he showed hyperleukocytosis; the hemogram profile revealed a white blood cell (WBC) count of 25.9 × 10^9^/L, consisting of 82% neutrophils, 11% lymphocytes, 4% monocytes, 2% basophils, 1% eosinophils, a hemoglobin level of 15.1 g/dL, and platelet count of 800 × 10^9^/L, and the physical examination was negative and no hepatosplenomegaly was observed. After exclusion of infectious origin, he performed second-level exams. The BM smear showed increased cellularity, with a high number of megakaryocytes without blast cells (Figure 1). The BM biopsy revealed a marrow cellularity of 90%, with markedly increased granulocytic series, normal megakaryocytes and reticulin staining. The karyotype was 46, XY, t (9;22) (q34;q11) in all the 10 metaphases analyzed. The real-time quantitative polymerase chain reaction showed the presence of the transcript p210 b3a2. Analysis of JAK2 V617F revealed a positivity of 0.7%, and the screening for CALR and cMPL mutations was negative. Therefore, a diagnosis of chronic-phase CML was made, with low Sokal (0.73), Hasford (695.51) and EUTOS (14) scores, and an intermediate ELTS (2.1065) score. The patient started treatment with imatinib, without cytoreduction and with hydroxyurea.

In the first phase, a balance of the two treatments, imatinib and eltrombopag, was necessary to normalize the platelet count. Eltrombopag was initially discontinued and then when the platelet count fell to 60 × 10^9^/L, it was restarted at 25 mg every other day for the first week, and then a daily dose of 25 mg was able to reach and maintain a platelet count of around 150 × 10^9^/L. The profile of WBC and platelets from diagnosis to complete hematological response with treatment is shown in Table 1. The patient continued treatment with imatinib and eltrombopag, and he reached cytogenetic remission and major molecular remission at 6 months. Following cytogenetic analysis, he showed a normal karyotype (46,XY), except for a deletion of chromosome Y in 3 cells out of 20 in September 2020. Since May 2018, the patient showed molecular remission (sustained MR ≥4, at last follow-up in June 2021 MR 5). Molecular analysis confirmed a low positivity for JAK2 (2.7%) in September 2017, which was reduced to 0.02% in December 2017, and then it remained negative (0%) thereafter. The results of the molecular analysis of bcr-abl and JAK2 are shown in Figure 2. No adverse events were observed during the follow-up at 4 years. The platelet count was in the normal range, with eltrombopag 25 mg/day, until October 2021 when we tried to suspend it; the first week after the discontinuation of eltrombopag, the platelet count reduced, and in the following week it failed at 51 × 10^9^/L. Therefore, the patient had to restart therapy with eltrombopag at the same dosage of 25 mg/day, obtaining normalization of the platelet count in two weeks this way.

## 3. Discussion

No relevance between autoimmune disease, such as ITP, and the development of CML has been noted to date. In the literature, we did not find case reports similar to this, except for two pediatric cases. The first was described in 1981; in this case report, an 8-year-old girl was diagnosed with ITP, which later became refractory to steroids, splenectomy, and immunosuppressive therapy, while she was being treated for pneumococcal meningitis. At 15 years old, she was found to have Philadelphia chromosome-positive CML and she died at 18 years of age [5]. The second pediatric case, published in 2003, concerned a 32-year-old male who developed CML 29 years after the diagnosis of ITP. He was treated with steroids successfully; however, at relapse, he was unresponsive to intravenous immunoglobulins and he underwent splenectomy. He was also treated with interferon-alpha for hepatitis C, presumably acquired from the immunoglobulins. The suboptimal platelet count normalized in 2001 when the patient also showed an increase in WBC. His BM smear showed typical characteristics of CML, and the diagnosis was confirmed by FISH and cytogenetics. The patient treated with imatinib 400 mg/day reached molecular and cytogenetic remission [6]. As far as adult case report concerns, there is a description, published in 2014, of a 77-year-old Japanese man with ITP treated with eltrombopag, because thrombocytopenia was refractory to prednisolone, which developed into CML. The patient was treated for CML with dasatinib, but a blastic crisis occurred and he died, despite switching to nilotinib. The authors noted the possible adverse event likely occurred due to the long-term use of eltrombopag; this was the first report of a TPO receptor agonist possibly contributing to the onset of CML and the transformation into a blastic phase [7]. No other experiences of patients having therapy-related CML (t-CML) in conjunction with chronic ITP have been published. Reports of t-CML were published in association with other hematological neoplasms, such as Hodgkin lymphoma and chronic lymphocytic leukemia [8,9].

In our case, we recently considered the possibility of treatment-free remission [10,11,12]; we tried with eltrombopag first, which was unsuccessful, but this does not mean that TFR cannot be tried with imatinib.

This report also allows us to reflect on the diagnostic process with the strict monitoring of blood values, due to regular visits for ITP, and the role of BM smear as a first approach rather than more detailed diagnostic exams, such as FISH, cytogenetics, and, more recently, molecular biology. 

The role of low allele burden JAK2 V617F is intriguing, but its significance remains obscure.

## 4. Conclusions

In this paper, we reported the unique case of CML diagnosed in an ITP patient, in whom treatments with imatinib and eltrombopag were feasible and well tolerated, allowing molecular remission of bcr-abl to be reached and stabilizing the platelet count.

## Figures and Tables

**Figure 1 medicina-57-01326-f001:**
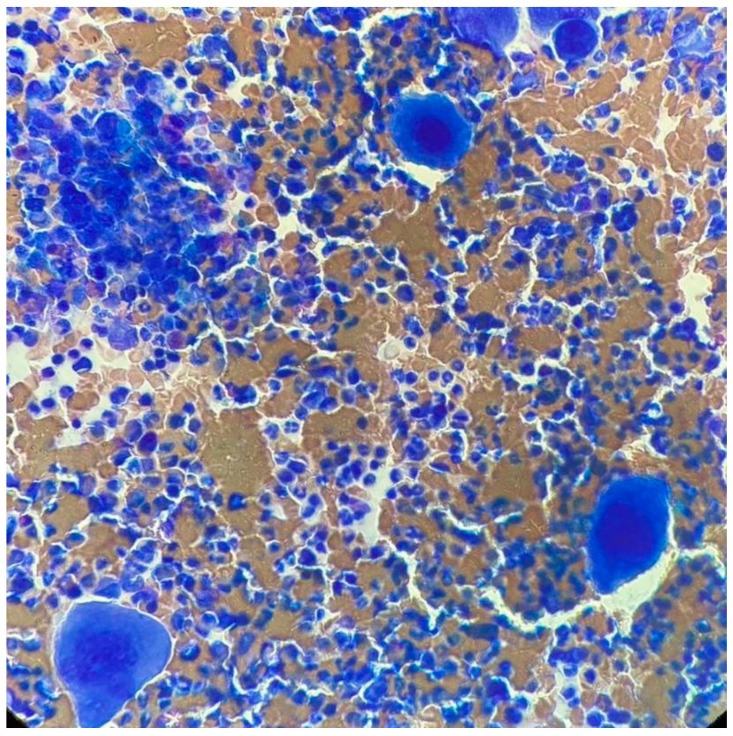
Bone marrow smear upon CML diagnosis.

**Figure 2 medicina-57-01326-f002:**
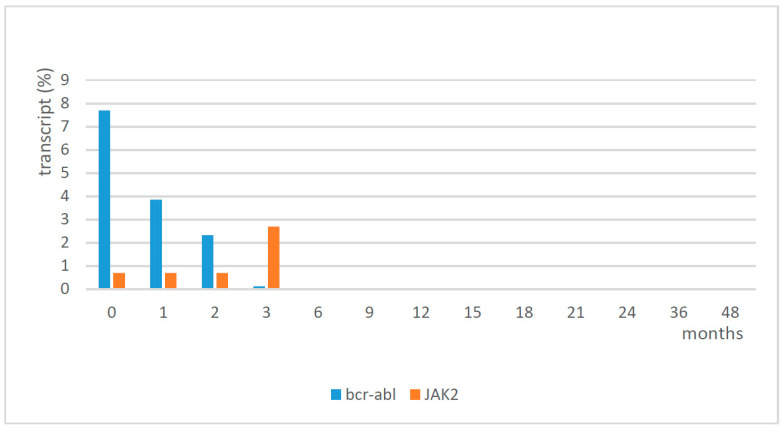
Molecular analysis of bcr-abl and JAK2.

**Table 1 medicina-57-01326-t001:** Profile of white blood cells and platelet count.

Days from Diagnosis	White Blood Cell(10^9^/L)	Neutrophil Count(10^9^/L)	Platelet Count(10^9^/L)	Treatment
0	25.9	21.3	800	Start imatinib, stop eltrombopag
7	21.8	17.9	620	
21	6.2	5.1	160	
35	5.0	3.2	60	Start eltrombopag 25 mg every other day
42	4.1	3.1	73	Eltrombopag 25 mg/day
49	4.2	3.0	149	
60	4.4	3.1	167	
90	4.1	2.9	156

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
