# Peer review of "Chronic Myeloid Leukemia in a Patient with Previous Idiopathic Thrombocytopenic Purpura: How to Manage Imatinib Together with Eltrombopag"

_medicina, 2021, doi:10.3390/medicina57121326_

Round 1

Reviewer 1 Report

This is a case report about a patient with long-standing ITP treated with eltrombopag who developed hyperleukocytosis during follow-up; after specific laboratory exams it was diagnosed as CML and he added imatinib for leukemia.

Authors conclude that imatinib and eltrombopag were well tolerated and effective.

Although the case report presents a rare co-occurrence, the significance of content is low and overall the report is not interesting.  

Author Response

We are re-submitting to your attention our revised case report entitled “Chronic myeloid leukemia in a patient with previous idiopathic thrombocytopenic purpura: how to manage imatinib together with eltrombopag”.

We appreciate the time and dedication of the reviewers and the editor and their constructive criticism.

We revised the discussion and conclusion section according to reviewers’ suggestions.

We improved the discussion with more references and moved the citations from conclusion to discussion.

We added the third case as suggested (Med Pediatr Oncol. 2003 Aug;41(2):159-60).

We made a Table demonstrating blood profile of patient at diagnosis with ITP and CML, and after treatment showing complete hematological response and we added a second Figure about bone marrow smear.

We consider this case report interesting and unique because we could use imatinib and eltrombopag together, they were well tolerated, allowing to reach molecular remission of bcr-abl and stabilizing platelet count. In comparison to other case reports in which eltrombopag was not available and more limitations associated with therapy were present.

We also think that this paper could be helpful to clinicians involved in the treatment of patients with chronic myeloid leukemia and idiopathic thrombocytopenic purpura.

We hope that the revised version will be considered acceptable for publication in Medicine.

Reviewer 2 Report

  1. The discussion needs to be expanded - points to consider are autoimmunity and leukaemias/ development of CML following ITP ( there is one more case-Med Pediatr Oncol. 2003 Aug;41(2):159-60), therapy-related CML, concomitant use of TKI and eltrombopag and use of eltrombopag in managing thrombocytopenia associated with TKI therapy in patients with CML
  2. Need English language editing

Author Response

(The authors gave the same response as above.)

Reviewer 3 Report

In this case report by Autore et al., the authors have described the clinical management of MDS/CML patient’s pre-diagnosed with idiopathic thrombocytopenic purpura. Citing recent advancement in the field the authors suggested that imatinib and eltrombopag combination therapy is well tolerated in these patients and allowed to reach complete molecular remission of CML as well as of ITP. The topic is of interest for both researchers and clinicians.

Major comments:

  1. The authors should add a table demonstrating blood profile of patient at diagnosis with ITP and CML, and after treatment showing complete hematological response.
  2. The authors should add figures representing the BM biopsy staining and karyotype analysis.
  3. In section 2, case report the authors have mentioned Figure 1, but no figure was attached. The authors should check the same.
  4. The discussion section is very weak. Citing previous studies (PMID: 27597907, 24179370, 12825226, and 12505128 etc) the authors should discuss the clinical spectrum, clinical management and outcome in ITP followed by MDS/CML patients.
  5. Also in the conclusion section the authors should include the limitations associated with current therapy and future directions for diagnosis and timely management of ITP associated MDS/CML patient’s.

Author Response

(The authors gave the same response as above.)

Round 2

Reviewer 1 Report

Patients with ITP have been found to have increased risk of hematological malignancies, including Hodgkin’s lymphoma, non-Hodgkin’s lymphoma and myeloproliferative neoplasms. This is a case report on the occurrence of chronic myeloid leukemia (CML) in a patient with idiopathic thrombocytopenic purpura (ITP).  The authors briefly described 65 years-old gentlemen who was diagnosed with CML following 15 years of follow-up for ITP. The authors also reviewed the literature and discussed two similar pediatric cases and one adult case in discussion section.

Given the rarity of myeloproliferative neoplasms, there are only sparse data available regarding the coexistence of ITP and CML. I believe, this case report will add valuable information to the available literature.

Please find below some minor suggestions;

1- A recent paper examined the possible link between ITP and the risk of overall and organ-specific cancers in Swedish population. They found that compared with a matched cohort without ITP, patients with ITP had a higher risk of developing malignancy, highest risk occurring during the first year post-diagnosis. The risk also found to have increased for myeloid malignancies (Charlotta Ekstrand et al, ‘’Cancer risk in patients with primary immune thrombocytopenia-A Swedish nationwide register study’’, published in Cancer Epidemiology in 2020).

Another paper investigated the risk of myeloid malignancies in patients with autoimmune conditions. Although they did not include ITP in the study, overall autoimmune conditions were not found to be associated with CML or MPN (Anderson LA et al, ‘’ Risks of myeloid malignancies ,n patients with autoimmune conditions’’, published in British Cancer Journal in 2009). That might also be a result of rarity of MPNs.

I believe, including information from literature regarding the coexistence of MPNs with ITP, as such above, would add value to the paper. However I am leaving this to the authors.

2- ITP is now more commonly referred as immune thrombocytopenia. I suggest using immune thrombocytopenia instead of idiopathic thrombocytopenia.

Thank you very much once again for including me as a reviewer for this paper.

Reviewer 3 Report

The authors have significantly modified the case report. I have one minor suggestion regarding Figure 2. It would be great if the authors describe the method of Real time PCR and Primer/Probe used in the figure 2 legend.